# Gender Differences in Protein Consumption and Body Composition: The Influence of Socioeconomic Status on Dietary Choices

**DOI:** 10.3390/foods14050887

**Published:** 2025-03-05

**Authors:** Mauro Lombardo

**Affiliations:** Department for the Promotion of Human Science and Quality of Life, San Raffaele Open University, Via di Val Cannuta, 247, 00166 Rome, Italy; mauro.lombardo@uniroma5.it

**Keywords:** gender differences, protein consumption, body composition, fat mass, fat-free mass, socioeconomic status, dietary habits, meat intake, plant-based proteins, nutritional epidemiology

## Abstract

Introduction: Gender differences in eating habits and protein consumption patterns are determined by cultural, socio-economic, and behavioural factors. Men tend to consume more meat, while women prefer plant-based proteins; however, the impact of these choices on body composition remains unclear. Objectives: This study explores gender differences in protein consumption and the influence of socioeconomic status on dietary choices, evaluating their potential association with body composition parameters, including fat mass (FM%) and fat-free mass (FFM%). Methods: A cross-sectional study was conducted on 1708 Italian adults (721 males, 987 females). Food intake was assessed using a validated 7-day food diary. Participants were classified as non-consumers, low consumers, or high consumers according to the median intake of each protein source. Body composition was measured using bioelectrical impedance analysis. Results: Meat and processed meat consumption was significantly higher in men (*p* < 0.001), while women consumed more soy (*p* = 0.0087). Individuals with high meat and processed meat consumption had a higher BMI (*p* < 0.01), whereas soy consumption was associated with a lower BMI (*p* = 0.0027). Socioeconomic status influenced food choices: low-income men consumed more meat and processed meat compared to higher-income groups (*p* < 0.01), while differences in fish consumption were mainly observed in low-income groups. Conclusions: Gender differences in protein consumption persist across socioeconomic levels and are related to body composition. Meat consumption is culturally linked to masculinity, whereas plant-based proteins are more commonly consumed by women. Understanding these dietary patterns can inform targeted nutritional interventions that promote balanced and sustainable diets.

## 1. Introduction

Understanding the relationship between gender, socioeconomic status, and dietary protein consumption is essential for developing effective nutritional interventions. Studies consistently report significant differences in dietary patterns: men tend to consume more meat, while women prefer plant-based proteins [1,2]. These choices are influenced by cultural norms, economic factors, and health perceptions rather than physiological needs alone [3].

Gender differences in protein consumption also extend their impact on body composition, as dietary habits can influence fat mass (FM) and fat-free mass (FFM). While previous research suggests that meat consumption is culturally associated with masculinity [1] and that plant-based protein intake is more prevalent among women due to ethical and health considerations [2], the direct effects of these choices on body composition remain unclear. Furthermore, socioeconomic status plays a crucial role in shaping food choices, with low-income individuals often relying on more accessible but less nutritionally diverse protein sources [4]. Studying these associations is essential to understand how dietary habits contribute to anthropometric outcomes and to inform targeted nutritional interventions that promote balanced and sustainable diets.

Socioeconomic status influences food choices, affecting both the availability and selection of protein sources [4]. Individuals with higher socioeconomic status generally have greater financial resources and nutritional awareness, which allow for a more diversified protein intake, including lean meats and plant-based alternatives [5,6]. In contrast, those in lower socioeconomic groups may rely more on processed meats and energy-dense foods due to accessibility and cost constraints [7].

Gender-specific eating patterns also vary across cultures. In many Western societies, meat consumption is associated with masculinity and strength [8], while in Asian countries such as Japan and South Korea, plant proteins, particularly legumes and soy, are widely consumed by both sexes [9]. In some African populations, food choices are driven more by economic factors and local traditions than by gender differences [10]. Beyond cultural influences, the switch to plant-based diets is often motivated by health, environmental, and ethical considerations [11]. In Northern Europe, for example, increased environmental awareness has led to an increase in the consumption of alternative proteins among genders [12]. Ethical concerns related to animal welfare further contribute to gendered eating behaviour, as women are more likely to reduce consumption of animal products for moral reasons [13,14,15].

This study investigates how gender and socioeconomic status influence protein consumption patterns and their potential association with body composition. The primary outcome is the relationship between protein intake and FM and FFM levels. Secondary outcomes include the persistence of gendered dietary habits across income levels and the influence of socioeconomic factors on protein choices. By considering both gender and income level, this study provides a broader perspective on the complex interactions between diet, socioeconomic status, and body composition.

## 2. Methods

### 2.1. Study Design and Participants

This cross-sectional study was conducted at a specialised medical centre in Rome, Italy, and focused on nutrition and metabolic health. Recruitment took place between January 2023 and November 2024, with data collected through structured dietary assessments and body composition measurements. Participants had to be adults (≥18 years), speak fluent Italian, and be able to complete a detailed dietary survey. Written informed consent was obtained prior to participation. Participants received standardised portion size guidelines to improve the accuracy of dietary data. Exclusion criteria included missing or incomplete dietary or anthropometric data as well as extreme or implausible body composition values. Extreme or implausible values were excluded on the basis of biological plausibility. The presence of metabolic disorders was determined through a combination of medical history and medical records, when available. A power analysis was conducted to determine the minimum sample size required to detect significant differences in fat mass percentage (FM%) between dietary protein intake groups, adjusted for age and gender. Based on an expected effect of η^2^ = 0.02, a power of 0.80, and an alpha level of 0.05, a sample size of approximately 2000 participants was initially estimated. Following the application of exclusion criteria—removing 292 participants due to missing or incomplete answers, implausible anthropometric values, or diagnosed metabolic disorders—the final sample included 1708 participants. Given that the reduction in sample size was approximately 14.6% of the estimated requirement, the statistical power of the study remains sufficiently robust for detecting meaningful differences. Post hoc power calculations confirmed that the final sample size maintained a power level above 0.78, ensuring the validity of the analyses. The study was approved by the Lazio Area 5 Territorial Ethics Committee (Approval Code: N.57/5R/23) and registered on ClinicalTrials.gov (NCT06654674).

### 2.2. Dietary Assessment

Food intake was assessed using validated 7-day food diaries, in which participants recorded their habitual consumption of meat, processed meat, fish, eggs, dairy products, legumes, and soy. Participants received standardised guidance on portion estimation before completing the diaries. Registered dieticians manually reviewed each diary, extracting detailed information on the frequency and portion sizes of each protein source. Participants were classified as low or high consumers according to the median intake of total protein, animal protein and vegetable protein. The median was calculated for the entire sample. Although men generally consume higher absolute amounts of protein, this approach allows a direct comparison of dietary intakes without imposing gender-based differences. However, to account for gender-related variations, gender was included as a covariate in all statistical analyses. Those who declared no intake were classified as non-consumers, while the remaining participants were divided into low and high consumers according to the median intake among consumers. The use of 7-day food diaries is subject to possible recall and social desirability bias. To mitigate these effects, participants received detailed instructions on how to fill out the diary, with reference materials for standard portions. Registered dietitians reviewed the diaries and addressed inconsistencies. Previous studies have validated this methodology by demonstrating concordance with biomarkers of protein intake [16,17]. Recruitment spanned 22 months to account for seasonal variability, although dietary changes across seasons were not specifically monitored. A study by Brunner et al. [16] compared a 7-day food diary with a food frequency questionnaire, validating both against biomarkers. Results indicated that the food diary provided reasonable agreement with biomarkers for energy and macronutrient intake, including protein, suggesting its utility in accurately capturing protein consumption. Furthermore, the Malmö Food Study [17] assessed the relative validity of dietary assessment methods by comparing a food frequency questionnaire and a food diary with an 18-day weighed food record. The results showed that the food diary method produced fairly good correlations with the reference method for most nutrients, including protein, after energy adjustment. Despite their limitations, 7-day food diaries were selected for their ability to provide detailed intake data over a short period. In this study, the term plant protein refers exclusively to legumes and soy, as these were the only plant-based sources recorded in the food diaries. Other plant protein sources, such as cereals, pseudocereals, nuts, and seeds, were evaluated only in terms of preference and were not included in the quantitative dietary intake analysis.

### 2.3. Body Composition Measurements

Anthropometric and body composition assessments were conducted following standardised procedures. Measurements were performed on an empty stomach, with participants wearing only undergarments, to minimise variability due to hydration status. Body weight was recorded with a calibrated electronic scale (Tanita BC-420 MA, Sportlife, Tokyo, Japan), with an accuracy of 100 g and a measurement range of 0–200 kg. Participants stood still on a hard, unpaved surface while a clinical assistant recorded their weight to the nearest gram. Height was measured with a stadiometer, with participants standing barefoot, with their heels, buttocks, shoulders, and head aligned with the vertical bar. The body mass index (BMI) was calculated as weight (kg) divided by height squared (m^2^). Abdominal circumference (AC) was measured at the midpoint between the iliac crest and the lowest rib, with participants standing and during minimal breathing. Body composition parameters, including FM, FFM, were assessed using the Tanita BC-420 MA Bioelectrical Impedance Analyser (BIA), a validated instrument compared to the BodPod [18]. Pre-assessment guidelines required participants to fast for at least three hours, to have abstained from strenuous physical activity for at least 12 h, and to have avoided excessive consumption of food, alcohol, and caffeine in the 12 h prior to measurement. To reduce variability due to fluid retention, female participants were assessed outside the luteal phase of the menstrual cycle, which is characterised by increased water retention and potential fluctuations in body composition measurements. This approach aimed to improve the accuracy of FM% and FFM% estimates. In addition to the body mass index, we calculated the Body Shape Index (BSI), a parameter that integrates height, BMI, and waist circumference to provide a more accurate measure of cardiometabolic risk. The BSI was calculated using the formula proposed by Krakauer and Krakauer [19], which considers waist circumference in metres divided by BMI raised two-thirds and height raised one-half. Higher values indicate a greater distribution of abdominal fat and an increased cardiometabolic risk, even in individuals with a normal BMI.

### 2.4. Physical Activity Assessment

Physical activity was assessed by means of a structured questionnaire collecting information on the type of sport practised and the amount of hours per week devoted to physical activity. Participants were divided into three physical activity categories: <5 h per week, 5–10 h per week, and >10 h per week. The questionnaire was discussed individually with a dietitian experienced in sports nutrition, who is also trained as a trainer. This comparison made it possible to validate the consistency of the answers and to correct any discrepancies.

### 2.5. Statistical Analysis

Continuous variables were tested for normality and presented as mean ± standard deviation (SD). Categorical variables were expressed as percentages. Differences between males and females were assessed using Welch’s *t*-test for continuous variables and chi-square test for categorical variables. To assess the relationship between dietary protein intake and body composition (BMI, FM%, FFM%), participants were classified into non-consumers, low consumers (below the median among consumers), and high consumers (above the median among consumers) for each food source. One-way ANOVA was used to compare differences between these groups, with Bonferroni post hoc adjustments when necessary (*p* < 0.05). Independent t-tests were used for the comparison between two groups. The Mann–Whitney U-test was used to assess gender differences in food consumption between the various protein sources. Kruskal–Wallis tests were performed to assess variations in food consumption between income levels within the FM% and FFM% groups, followed by post hoc Dunn tests where applicable. SPSS version 23.0 (SPSS Inc., Chicago, IL, USA) and Python (version 3.12) were used. A significance level of *p* < 0.05 was considered statistically significant and results were reported with exact *p*-values where applicable.

## 3. Results

The study population comprised 1708 participants, of whom 721 were male (42.2%) and 987 female (57.8%). Baseline characteristics, stratified by gender, are summarised in Table 1. Males and females showed significant differences in several anthropometric measures. Males had a higher weight (88.9 ± 17.7 kg versus 72.7 ± 14.5 kg, *p* < 0.001), FFM% (71.4 ± 7.4% versus 62.1 ± 7.3%, *p* < 0.001), and basal metabolic rate. In contrast, females had a significantly higher FM% (34.7 ± 7.6% vs. 24.8 ± 7.8%, *p* < 0.001) and BMI. Abdominal circumference was greater in males. The mean BSI values differed significantly between men and women. Men had a higher mean value than women, suggesting a different distribution of body mass and abdominal fat. With regard to lifestyle factors, 22.6% of participants were regular smokers, with a slightly higher prevalence in females (23.2% vs. 21.9%, *p* < 0.001). Physical activity levels were significantly different between genders, with 61.0% of males regularly practising sport compared to 49.7% of females (*p* < 0.001). Males also reported spending more time on sporting activities, with 23.7% exercising for 5–10 h per week compared to 11.8% of females (*p* < 0.001) and 3.9% exercising for >10 h per week compared to 1.7% of females. Socioeconomic factors showed significant differences between genders. A higher percentage of females were in the lowest income bracket (<20,000 euro; 20.1% vs. 17.3%, *p* < 0.001), whereas the income distribution in the other brackets was comparable between the genders. The occupational categories also varied: males were more frequently employed in professional services (21.9% vs. 14.2%), while females were over-represented in health and care (8.9% vs. 4.9%, *p* < 0.001).

Figure 1 shows the distribution of dietary protein intake among non-consumers, low consumers, and high consumers. Soy intake was the lowest, with only 2.5% (*n* = 45) classified as high consumers, while 85.5% (*n* = 1545) were non-consumers. Legumes were more evenly distributed, with 21.3% (*n* = 384) being high consumers and 24.9% (*n* = 446) non-consumers. Proteins of animal origin showed a wider intake. Meat and fish recorded the highest percentages of high consumers (46.5% and 43.0%), while processed meat recorded the highest percentage of non-consumers (59.2%). The consumption of eggs and dairy products followed a mixed pattern, with dairy products having the highest percentage of low consumers (60.2%) and eggs having the highest percentage of non-consumers (47.0%).

As shown in Figure 2, significant correlations were observed between BMI and the different dietary protein sources. High meat consumers had a mean BMI of 28.34 ± 5.49 kg/m^2^, whereas non-consumers had a mean BMI of 26.93 ± 4.94 kg/m^2^ (*p* = 0.0027). Soy intake was correlated with a lower BMI, with a mean BMI of 25.59 ± 4.73 kg/m^2^ (*p* = 0.0027). High egg consumers had a mean BMI of 26.90 ± 4.93 kg/m^2^ (*p* = 0.0047). Processed meat and dairy products showed a positive correlation with BMI. High consumers of processed meat had a mean BMI of 28.88 ± 5.76 kg/m^2^ (*p* = 0.0046), while high consumers of dairy products had a mean BMI of 28.67 ± 5.62 kg/m^2^ (*p* = 0.0247). Legumes and fish were not significantly correlated with BMI (*p* = 0.9225 and *p* = 0.3275, respectively).

Figure 3 illustrates the relationship between dietary protein sources and FM%. High meat consumers had a mean FM% of 29.78 ± 9.60%, while non-consumers had a mean FM% of 30.26 ± 8.84%. High consumers of fish had a mean FM% of 29.54 ± 9.04%, while non-consumers had a mean FM% of 30.74 ± 9.31%. Similarly, high consumers of eggs had an FM% of 27.51 ± 9.20%. Dairy intake was positively correlated with FM%, with a mean FM% of 30.56 ± 9.81% for high consumers of dairy products. The intake of processed meat, legumes, and soy was not significantly correlated with FM%.

As shown in Figure 4, FFM% was significantly correlated with multiple food sources. Egg and soy high consumers showed the highest mean FFM%. Meat and fish intake also showed a positive correlation, with high consumers showing mean FFM% values of 66.70 ± 9.16% (*p* = 0.008) and 66.86 ± 8.71% (*p* = 0.00479), respectively. Dairy consumption was positively correlated with FFM%, with high consumption showing a mean of 66.07 ± 9.31% (*p* = 0.00016). Processed meat and legumes were not significantly correlated with FFM% (*p* = 0.4212 and *p* = 0.2179, respectively).

Figure 5 presents the significant differences in average food consumption between men and women in four categories according to FM% and FFM%. The dataset included 853 participants (360 men, 493 women). Classification was based on the median of FM% and FFM% within each sex: high FM% was defined as >25.4% for men and >38.6% for women, while low FM% was ≤25.4% for men and ≤38.6% for women. High FFM% was >73.2% for men and >59.4% for women, while low FFM% was ≤73.2% for men and ≤59.4% for women. Among individuals with high FFM%, men consumed significantly more meat (*p* = 0.000005), processed meat (*p* = 0.0022), and fish (*p* = 0.000001) than women. Similar trends were observed in the high FFM% group, where men reported a significantly higher consumption of meat (*p* = 2.7 × 10^−8^), processed meat (*p* = 0.0006), and fish (*p* = 1.1 × 10^−5^). The same pattern was found in subjects with low FFM%, where men consumed more meat (*p* = 1.68 × 10^−6^), processed meat (*p* = 0.0015), and fish (*p* = 6.20 × 10^−7^) than women. Women in the low FFM% group had a significantly higher intake of soya (*p* = 0.0087).

Men consumed more meat than women in all income categories (Figure 6), with significant differences observed in the <20,000 euro group (*p* = 1.49 × 10^−7^), the 20,000–40,000 euro group (*p* = 7.07 × 10^−6^), the 40,000–60,000 euro group (*p* = 7.81 × 10^−4^), and the >60,000 euro group (*p* = 0.0168). With regard to fish consumption, men consumed significantly more fish than women in the lowest income category (*p* = 0.0010), whereas no significant differences were found in the highest income groups.

In addition to actual food intake, the participants’ food preferences were explored by means of a short questionnaire assessing attitudes towards different plant protein sources. Although legumes and soy were the main plant protein sources considered in this study, additional data were collected on cereals and nuts. However, statistical analyses revealed no significant associations between gender or income and preference for these foods (Appendix A).

## 4. Discussion

### 4.1. Gender Differences in Protein Consumption and Body Composition

Gender differences in protein consumption are shaped by cultural, social, and psychological factors. Our study confirms that men consume more meat and processed meat than women, while the latter prefer plant-based proteins, particularly soy. These findings align with those of De Backer et al. [1], who highlighted the cultural association of meat with masculinity, reinforcing gender identity. Similarly, Nezlek and Forestell [2] observed that vegetarianism is predominantly driven by women, suggesting that the reduction in meat consumption is more related to cultural factors than physiological needs. However, our study expands this perspective by showing that these dietary preferences also translate into significant differences in body composition, with men exhibiting higher FFM% and women a higher FM%. This observation supports our previous findings [14], which reported an interplay between diet, physical activity, and body composition across genders. Soy consumption, more frequent among women with lower FM%, further supports the notion that dietary choices are associated with body composition. Our results are consistent with Yuan et al. [20], who found that diets richer in plant-based proteins are associated with a lower BMI, particularly among women. Additionally, the role of soy in hormonal balance has been well documented: Rizzo et al. [21] emphasised its benefits in female hormonal regulation, while Hamilton-Reeves et al. [22] disproved concerns regarding its negative impact on testosterone levels in men. The cultural perception of soy as a “feminine” food may contribute to its lower acceptance among men, as also noted by Adamczyk et al. [23]. Moreover, our study found a positive association between meat consumption and FFM% in men, but no such link with processed meat. This contrasts with the findings of Camilleri et al. [8], which suggested that high-protein diets, including those containing processed meats, contribute to muscle growth. This discrepancy may stem from differences in study populations, as previous research often focused on athletes or individuals following structured high-protein diets [24], whereas our study examined the general population. Additionally, while Evans et al. [25] demonstrated that protein intake helps preserve muscle mass during weight loss—particularly in men—our findings suggest that protein quality and food sources play a more critical role than just total intake.

### 4.2. Socioeconomic Influence on Protein Choices

Our study highlights the strong influence of socioeconomic status on protein consumption, particularly among men. We observed that lower-income men consume more meat and processed meat, while lower-income women are more likely to rely on plant-based protein sources. This pattern aligns with findings by Hosseinpour-Niazi et al. [26], who reported an association between low socioeconomic status and a higher intake of processed foods, including processed meat. Similarly, Rickerby et al. [27] found that higher-income individuals are more inclined to adopt plant-based diets, likely due to greater nutritional awareness and accessibility to diverse food options. However, in men, high meat consumption persists across all income levels, suggesting that cultural factors play a stronger role than economic constraints. Enriquez et al. [28] emphasised that food choices are not solely dictated by economic availability but are also shaped by social norms and cultural expectations. The preference for processed meat among lower-income men may be explained by its lower cost compared to fresh meat or fish. This is consistent with Pechey and Monsivais [7], who highlighted that individuals with lower economic means tend to prioritise affordability over nutritional quality when selecting protein sources. However, our findings diverge from Muhammad et al. [29], who observed a positive correlation between income and animal protein consumption in developing countries. In contrast, our data suggest that in Italy, processed meat remains a staple among lower-income groups, indicating a context-specific cultural preference rather than a simple economic constraint.

### 4.3. Fish and Dairy Consumption Across Income and Gender

Another key finding of our study is the higher fish consumption among lower-income men compared to women in the same income bracket, though this disparity narrows at higher income levels. This trend is consistent with Love et al. [30] who demonstrated that income and local food availability significantly impact fish consumption. However, while in the United States fish intake is heavily influenced by ethnicity and geographic factors, our study suggests that in Italy, cultural perceptions may be equally relevant. Regarding dairy consumption, our results show a positive correlation between dairy intake and BMI, independent of gender. This aligns with Michaëlsson et al. [31], who found that high milk consumption was linked to an increased risk of adiposity and mortality, challenging the traditional view of dairy as universally beneficial. However, unlike studies suggesting a stronger preference for dairy among women due to its perceived benefits for bone health [32], our findings indicate that gender differences in dairy consumption may be less pronounced in the Italian context.

### 4.4. Cultural Context and Implications for Public Health

Our study suggests that the Mediterranean cultural context may moderate some of the gender differences in protein consumption commonly reported in other countries [33,34]. The MD, characterised by a balanced intake of meat, fish, dairy, and legumes, may have mitigated disparities observed in Western countries with higher red meat consumption (e.g., North America) or those with strong vegetarian traditions (e.g., South Asia) [35,36]. Additionally, the deep-rooted presence of legumes [37] and fish [38] in Italian cuisine may explain why gender differences in protein consumption appear less pronounced compared to other cultural settings.

Furthermore, the role of women in meal preparation may contribute to greater dietary diversity and a stronger emphasis on perceived “healthier” protein sources. This is consistent with findings from Storz et al. [39] who noted that traditional gender roles influence food selection, with women often making choices aligned with health-conscious eating patterns. While these cultural influences reduce some gender disparities, they also reinforce traditional gendered eating behaviours, suggesting that public health interventions should address both economic and cultural barriers to dietary change.

### 4.5. Policy Implications and Future Directions

These findings underline the need for gender- and income-sensitive nutrition policies that promote balanced and sustainable protein consumption. Public health interventions should address misconceptions about protein sources and improve access to high-quality and diverse proteins for low-income groups. Specific strategies should consider differences in food choices, economic constraints, and cultural perceptions (Table 2).

This study has several limitations that must be considered. The use of self-reported dietary data, in particular, the use of 7-day food diaries, introduces potential recall and social desirability biases, which may have influenced the accuracy of food intake estimates. Although the participants received detailed instructions and the dieticians reviewed the records, these biases cannot be completely eliminated. The cross-sectional design prevents causal inference, making it unclear whether eating habits influence body composition or vice versa. Furthermore, body composition was assessed using BIA, a practical and widely used method, but one that has limitations compared to standard techniques such as DEXA. The study population, recruited from a single medical centre in Rome, may not fully reflect demographic and cultural variations in dietary habits, limiting the generalisability of the results. Seasonal variations in food intake, which may have influenced food consumption patterns, were not systematically monitored. Furthermore, unmeasured factors such as psychological influences, metabolic differences, and physical activity levels may have contributed to the observed associations. A detailed discussion of these limitations is available in the Appendix A.

## 5. Conclusions

Gender differences in protein consumption persist even when socio-economic factors are taken into account. Men prefer meat and processed meats, reinforcing cultural norms, while women show a greater preference for plant-based proteins, often linked to health consciousness. Socioeconomic status further modulates these choices: low-income men rely more on processed meats, while women in the same income bracket opt for plant-based alternatives. However, cultural influences play a decisive role beyond economic constraints, particularly in shaping men’s resistance to plant-based diets.

Our findings highlight the urgent need for targeted nutrition policies that go beyond generic dietary recommendations. There is a need to address misconceptions about protein quality, particularly among men, by reformulating plant-based protein as a performance enhancing rather than a restrictive factor. Similarly, interventions should ensure that low-income groups, regardless of gender, have access to diverse and high-quality protein sources. The Mediterranean dietary framework may offer a model for balancing these disparities, promoting protein diversity without reinforcing gendered dietary patterns.

## Figures and Tables

**Figure 1 foods-14-00887-f001:**
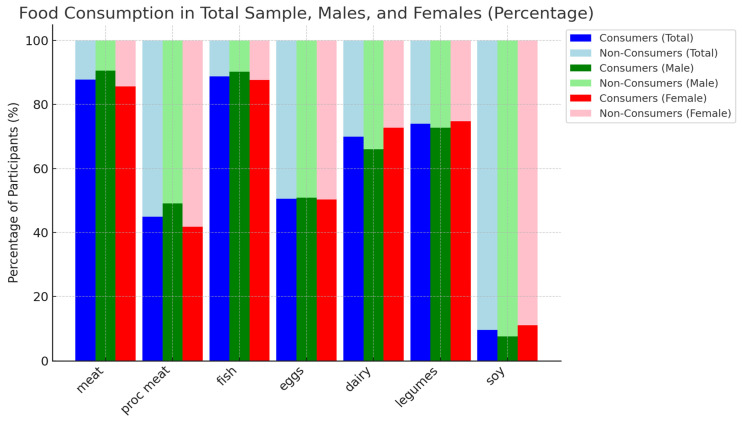
Consumption patterns of different food types among the total sample, males, and females, classified as consumers and non-consumers. Percentage of participants consuming each type of food in the total sample, and between males and females, classified as consumers and non-consumers. Consumers in the total sample are indicated in blue, while non-consumers in light blue. Male consumers are indicated in green, while non-consumers in light green. Female consumers are shown in red and non-consumers in pink. The figure shows the differences in consumption patterns of meat, processed meat, fish, eggs, dairy products, legumes, and soybeans among the population subgroups.

**Figure 2 foods-14-00887-f002:**
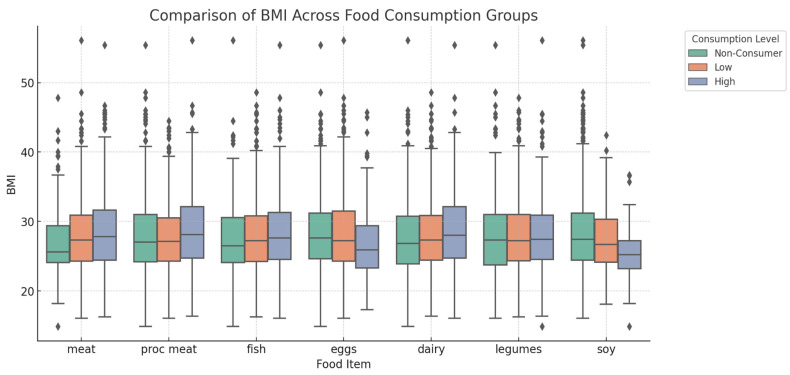
Correlation between dietary protein consumption and body mass index (BMI). Relationship between dietary protein consumption and body mass index (BMI). Participants were classified into non-consumers (intake of 0), low consumers (intake below the median among consumers), and high consumers (intake above the median among consumers) for each food source. Statistical analysis using one-way ANOVA revealed significant correlations for meat (*p* = 0.0027), processed meat (*p* = 0.0046), eggs (*p* = 0.0047), dairy products (*p* = 0.0247), and soy (*p* = 0.0027).

**Figure 3 foods-14-00887-f003:**
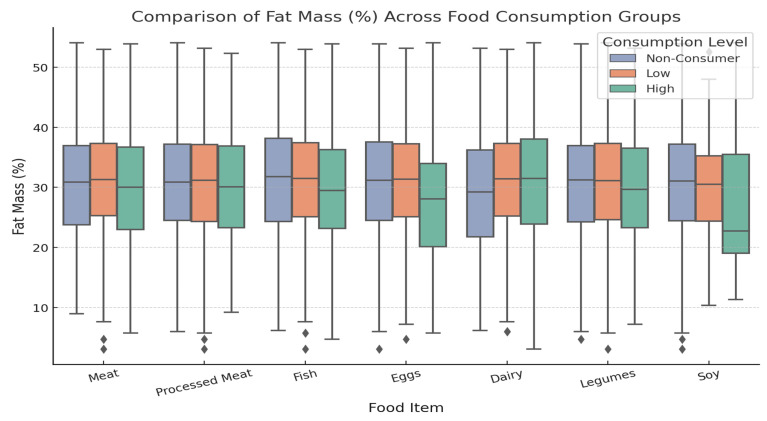
Correlation between dietary protein consumption and fat mass percentage (FM%). Relationship between dietary protein consumption and percentage of fat mass (FM%). Participants were classified into non-consumers (intake of 0), low consumers (intake below the median among consumers), and high consumers (intake above the median among consumers) for each food source. Statistical analysis using one-way ANOVA revealed significant correlations for meat (*p* = 0.007), fish (*p* = 0.002), eggs (*p* < 0.0001), and dairy products (*p* < 0.0001). Processed meat (*p* = 0.4438), legumes (*p* = 0.204), and soybeans (*p* = 0.718) were not significantly correlated with FM%.

**Figure 4 foods-14-00887-f004:**
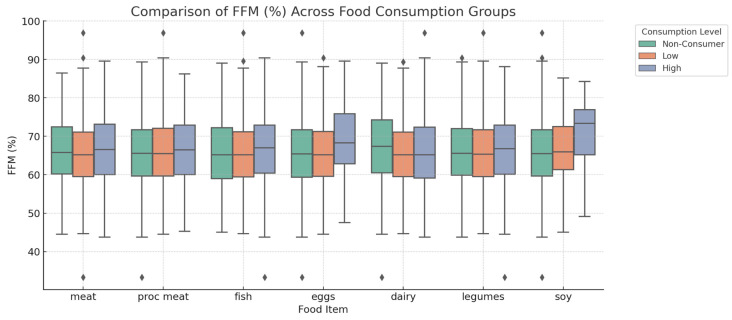
Correlation between dietary protein consumption and Fat-Free Mass Percentage (FFM%). Relationship between dietary protein consumption and percentage of fat free mass (FFM%). Participants were classified into non-consumers (intake of 0), low consumers (intake below the median among consumers), and high consumers (intake above the median among consumers) for each food source. Statistical analysis using one-way ANOVA revealed significant correlations for meat (*p* = 0.008), fish (*p* = 0.00479), eggs (*p* < 0.0001), dairy products (*p* = 0.00016), and soy (*p* = 0.0045). Higher consumption of meat, fish, eggs, and soy was correlated with a higher percentage of fat-free mass. Processed meat (*p* = 0.4212) and legumes (*p* = 0.2179) were not significantly correlated with FFM%. A *p*-value < 0.05 was considered statistically significant.

**Figure 5 foods-14-00887-f005:**
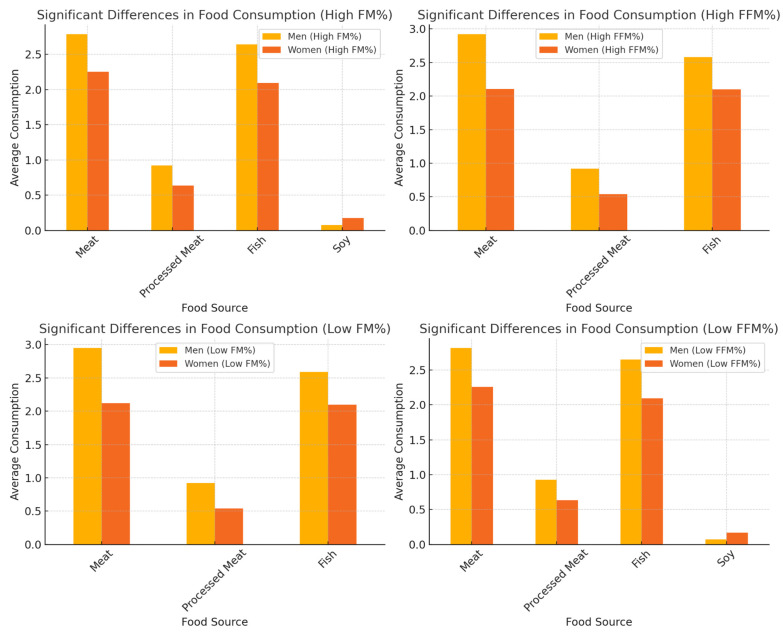
Gender differences in food consumption by fat mass and fat-free mass categories. This figure illustrates the significant differences in average food consumption between men and women in four groups classified according to percentage fat mass (FM%) and percentage fat-free mass (FFM%). The analysis was conducted using the Mann–Whitney U-test and only statistically significant differences (*p* < 0.05) were displayed. The dataset comprised 853 participants (360 men and 493 women). The classification was based on median values within each gender: for high FM%, the threshold was >25.4% for men and >38.6% for women, while for low FM% it was ≤25.4% for men and ≤38.6% for women. Similarly, for high FFM%, the threshold was >73.2% for men and >59.4% for women, while for low FFM% it was ≤73.2% for men and ≤59.4% for women.

**Figure 6 foods-14-00887-f006:**
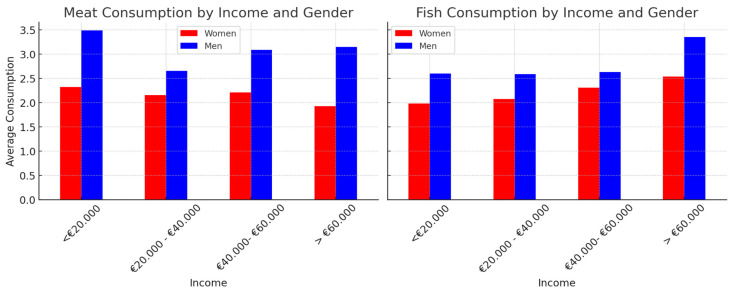
Meat and fish consumption by income and gender. This figure illustrates the differences in meat and fish consumption between men and women in the different income groups. The analysis was conducted using the Mann–Whitney U-test, with statistically significant differences (*p* < 0.05) indicated. The dataset comprised 1708 participants (721 men and 987 women), classified into four income groups: <20,000 euro, 20,000–40,000 euro, 40,000–60,000 euro, and >60,000 euro.

**Table 1 foods-14-00887-t001:** Descriptive statistics of anthropometric, lifestyle, and socioeconomic factors by gender.

Variable	Total (*n* = 1708)	Male (*n* = 721)	Female (*n* = 987)	*p*-Value	Cohen’s d
Age (years)	40.2 ± 14.2	39.2 ± 14.1	40.9 ± 14.3	0.019	
Current Smoker (%)	22.6%	21.9%	23.2%	<0.001	
Weight (kg)	79.5 ± 17.8	88.9 ± 17.7	72.7 ± 14.5	<0.001	1.013
BMI (kg/m^2^)	28.0 ± 5.2	28.7 ± 5.2	27.5 ± 5.2	<0.001	0.22
Body Shape Index (BSI)	0.0811 ± 0.0050	0.0815 ± 0.0044	0.0815 ± 0.0044	0.001	0.165
Fat Mass (%)	30.5 ± 9.1	24.8 ± 7.8	34.7 ± 7.6	<0.001	−1.288
Fat Mass (kg)	24.8 ± 10.7	23.0 ± 11.0	26.1 ± 10.4	<0.001	−0.287
Fat-Free Mass (%)	66.0 ± 8.7	71.4 ± 7.4	62.1 ± 7.3	<0.001	1.272
Fat-Free Mass (kg)	52.0 ± 11.3	62.5 ± 8.6	44.3 ± 5.3	<0.001	2.649
Abdominal Circumference (cm)	96.8 ± 14.3	101.1 ± 14.5	93.5 ± 13.2	<0.001	0.550
Basal Metabolic Rate (kcal/day)	1661.1 ± 349.1	1963.5 ± 282.2	1437.8 ± 188.8	<0.001	2.255
Income (%)					
<€20,000	323 (18.9%)	125 (17.3%)	198 (20.1%)	<0.001	
€20,000–€40,000	1135 (66.5%)	484 (67.1%)	651 (66.0%)		
€40,000–€60,000	202 (11.8%)	91 (12.6%)	111 (11.2%)		
>€60,000	46 (2.7%)	20 (2.8%)	26 (2.6%)		
Category Work (%)					
Sales and Services	785 (46.0%)	318 (44.1%)	467 (47.3%)	<0.001	
Professional Services	298 (17.4%)	158 (21.9%)	140 (14.2%)		
Healthcare and Wellness	123 (7.2%)	35 (4.9%)	88 (8.9%)		
Other	502 (29.4%)	210 (29.1%)	292 (29.6%)		
Do you play a sport? (%)					
Yes	931.0 (54.5%)	440.0 (61.0%)	491.0 (49.7%)	<0.001	
No	777.0 (45.5%)	281.0 (39.0%)	496.0 (50.3%)		
Sport hours per week (%)					
<5 h	687 (40.2%)	263 (36.5%)	424 (43.0%)	<0.001	
5–10 h	287 (16.8%)	171 (23.7%)	116 (11.8%)		
>10 h	45 (2.6%)	28 (3.9%)	17 (1.7%)		

Values are presented as mean ± standard deviation (SD) for continuous variables and as absolute numbers (percentages) for categorical variables. Group comparisons between males and females were performed using *t*-test for continuous variables and the chi-square test for categorical variables. The Body Shape Index (BSI) was calculated as waist circumference (AC) in metres divided by BMI raised to two-thirds and height raised to one-half, according to the formula BSI = AC/(BMI^(2/3) × Height^(1/2)) The *p*-values indicate the significance of the differences between genders and Cohen’s d values which quantify the effect size of gender differences for continuous variables. A *p*-value < 0.05 was considered statistically significant.

**Table 2 foods-14-00887-t002:** Gender-sensitive nutrition policies based on study findings.

Issue Identified	Policy Recommendation	Target Group
High meat and processed meat consumption in men	Promote plant-based protein sources through customised campaigns emphasising the benefits for health and muscle performance.	Men, particularly in lower-income groups
Socioeconomic disparities in fish consumption	Improving the accessibility and affordability of fresh fish in low-income areas; promoting its role in men’s dietary patterns.	Low-income populations, particularly men
Cultural perceptions influencing food choices	Develop communication strategies to counter gender-based food stereotypes and encourage balanced eating patterns.	General population
Gender-specific responses to nutritional messaging	Designing targeted interventions: performance and strength messages for men, longevity and sustainability for women.	Men and women
Misconceptions about plant-based proteins in men	Designing targeted interventions: performance and strength messages for men, longevity and sustainability for women.	Men, especially those engaged in physical activity

Caption Table 2—Summary of gender-related dietary issues identified in this study and proposed policy interventions. Strategies focus on addressing cultural perceptions, improving access to protein sources, and tailoring nutrition communication to different demographic groups.

## Data Availability

The data supporting the findings of this study are available from the corresponding author upon reasonable request. All data will be shared in a de-identified format to protect participant confidentiality.

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
