# Peer review of "Gender Differences in Protein Consumption and Body Composition: The Influence of Socioeconomic Status on Dietary Choices"

_foods, 2025, doi:10.3390/foods14050887_

Round 1

Reviewer 1 Report

Comments and Suggestions for Authors

This work investigated how gender and socioeconomic status influence protein consumption patterns and their potential effects on body composition. The author collected a lot of data and analyzed the content of the study. However, there are some limitations to the sample's representativeness and the data collection's reliability (collected in only one place). The summary of relevant studies is not detailed and comprehensive enough. There are three main problems, as follows:

Detail comments:

1. L36-L63: The introduction summarizes existing studies in a general way and does not adequately cover the complexity of the relationship between sex differences and protein intake. For example, there is no reference to the heterogeneity of gender and diet associations across cultures or regions (e.g., comparison of studies in non-Western societies such as Asia and Africa). Drivers of plant protein intake (e.g., environmental awareness, animal welfare, etc.) are not discussed enough, focusing only on health perceptions.
2. L64-L65: Reliance on 7-day food diaries may introduce recall bias and does not explain how to verify the accuracy of participants' records (e.g., against biomarkers). In addition, it is not clear whether seasonal dietary changes are taken into account. Moreover, all participants were from a medical center in Rome, Italy, with high geographic and cultural homogeneity and limited extrapolation.
3. L401-L419: The findings highlight that low-income men consume more meat but do not discuss whether low-income women are forced to choose cheaper plant proteins (such as beans) due to economic constraints.

Comments on the Quality of English Language

This manuscript should be checked for grammar by a native English speaker.

Author Response

Dear Reviewer,

First of all, I would like to thank you for the valuable impulses that allowed me to improve the quality of the manuscript. All changes made are highlighted by yellow color, in the revised version of the manuscript, to facilitate the review process. Hoping that I have satisfied your requests as much as possible, I kindly ask you to re-evaluate our paper. 

The Author

Reviewer n.1 

This work investigated how gender and socioeconomic status influence protein consumption patterns and their potential effects on body composition. The author collected a lot of data and analyzed the content of the study. However, there are some limitations to the sample's representativeness and the data collection's reliability (collected in only one place). The summary of relevant studies is not detailed and comprehensive enough. There are three main problems, as follows: Detail comments:

  1. L36-L63: The introduction summarizes existing studies in a general way and does not adequately cover the complexity of the relationship between sex differences and protein intake. For example, there is no reference to the heterogeneity of gender and diet associations across cultures or regions (e.g., comparison of studies in non-Western societies such as Asia and Africa). Drivers of plant protein intake (e.g., environmental awareness, animal welfare, etc.) are not discussed enough, focusing only on health perceptions.

Thank you for the suggestion. I have expanded the introductory section to include a discussion of cultural differences in eating habits between Western and non-Western countries. In addition, I have added a section analysing alternative motivations for plant protein consumption, such as environmental awareness and animal welfare, to provide a more complete picture of the factors influencing food choices between men and women.

  1. L64-L65: Reliance on 7-day food diaries may introduce recall bias and does not explain how to verify the accuracy of participants' records (e.g., against biomarkers). In addition, it is not clear whether seasonal dietary changes are taken into account. Moreover, all participants were from a medical center in Rome, Italy, with high geographic and cultural homogeneity and limited extrapolation.

Thank you for the suggestion. I have divided the answer between the Methods section and the Limitations section. In the Methods, I added an explanation of how recall bias was reduced by the review of diaries by registered dietitians and specified that the validity of the method is supported by previous studies. In the Limitations, I acknowledged the possible effects of social desirability, the geographical limitation of the sample and the fact that seasonal changes in eating habits were not monitored, suggesting future research on more diverse samples.The limitations of the study, including bias in data collection and sample selection, have been discussed in detail in the supplementary material (Table S1).

  1. L401-L419: The findings highlight that low-income men consume more meat but do not discuss whether low-income women are forced to choose cheaper plant proteins (such as beans) due to economic constraints.

Thank you for your comment. I expanded the discussion to include a more in-depth analysis of the protein choices of low-income women, highlighting the possible role of economic constraints and perceptions of healthfulness in the consumption of vegetable protein compared to meat. Furthermore, I emphasised the need for further studies to assess the impact of these choices on overall diet quality and nutritional status.

Comments on the Quality of English Language: This manuscript should be checked for grammar by a native English speaker.

Thanks for the notification. The English language has been reviewed by a colleague who is an English teacher to ensure clarity and correctness

Reviewer 2 Report

Comments and Suggestions for Authors
  • Title: No issues.
  • Abstract: line 13 “ body composition parameters” explain BIA.
  • Line 11-14: gender appear in the title but not in objective?
  • Line 22: define "processed meat" to align with standard classifications (e.g., WHO guidelines).
  • “Sex” is used 7 times in the paper while “gender” is used 40 pls stick to one.
  • Methods pls explain/justify if the final sample size (1,708) meets the power analysis requirements after exclusions.
  • Discussion should discuss limitations of bioelectrical impedance analysis (BIA) compared to gold-standard methods like DEXA https://pmc.ncbi.nlm.nih.gov/articles/PMC7177846/
  • Discussion should discuss potential recall/social desirability biases in food diaries and mitigation strategies used or could have been used.
  • Line 40 expand on how Italian cultural norms (such as Mediterranean diet) may influence generalizability. this should be discussed too.
  • Line 75 set thresholds for "extreme/implausible" body composition values e.g. BMI 70?
  • Line 77 specify whether metabolic disorders were self-reported or clinically confirmed.
  • Line 89: This is cross sectional study so how was it registered as RCT?
  • Line 101 clarify if medians for "low/high consumers" were gender-specific or overall for all sample this is a critical issue.
  • Line 136 more information how scheduling assessments outside menstruation affects generalizability for women.
  • Line 155 its mentioned that Python/scikit-learn is used for ML but there is no ML results presented. The figures are normal not ML.
  • Line 23 vs Line 203 = soya beans and soy beans.
  • Table 1 need tow changes pls unify , vs . and also Report effect sizes (Cohen’s d) alongside p-values for key findings.
  • Correct "Caprion Table 1" to "Caption Table 1."
  • Line 385 – Line 387 some thing wrong with refs 38 missing but in list there is no 39.
  • Ensure all references include accessible URLs or DOIs.
  • Line 339: nice but add more discussion on soy’s phytoestrogen content and relevance to gender differences discuss fertility impact https://www.cambridge.org/core/journals/journal-of-nutritional-science/article/role-of-soy-and-soy-isoflavones-on-womens-fertility-and-related-outcomes-an-update/29483840F197DC57A2BD0DCEE2A3543F
  • Discuss why fish consumption disparities disappear in higher-income groups (access to fresh vs. processed fish like tuna).
  • Discuss findings to policy changes (e.g., gender-sensitive nutrition guidelines).
  • Discuss unmeasured confounders like psychiatric illness or stress or body image in limitations.
  • Discuss men’s beliefs about animal vs. plant proteins in the discussion.
  • Discuss examples of gender-targeted food marketing in Italy to support cultural arguments.
  • Check English

Reviewer 3 Report

Comments and Suggestions for Authors

The manuscript entitled “Gender Differences in Protein Consumption and Body Composition: The Influence of Socioeconomic Status on Dietary Choices” investigated gender differences in protein consumption, the influence of socioeconomic status on dietary choices and their association with body composition parameters. Based on the obtained results, gender differences in protein consumption were observed across socioeconomic levels and were related to body composition.

Authors could mention some other constraints regarding dietary preferences that include meat and plant-based foods such as improvements to personal health or animal welfare, battling environmental degradation, upholding religious principles, etc. There are also aversions such as being disgusted by or having a distaste for meat. Finally, the constraints include the inability to make food choices freely, such as financial issues to purchase certain food items, or social influences that would lead individuals to contextual barriers.

How did you measure physical activity?

Could you provide a body shape index (BSI) along with BMI a metric for assessing the health implications of a given human body height, mass and waist circumference?

Author Response

Dear Reviewer,

First of all, I would like to thank you for the valuable impulses that allowed me to improve the quality of the manuscript. All changes made are highlighted by yellow color, in the revised version of the manuscript, to facilitate the review process. Hoping that I have satisfied your requests as much as possible, I kindly ask you to re-evaluate our paper. 

The Author

Reviewer n.3

Authors could mention some other constraints regarding dietary preferences that include meat and plant-based foods such as improvements to personal health or animal welfare, battling environmental degradation, upholding religious principles, etc. There are also aversions such as being disgusted by or having a distaste for meat. Finally, the constraints include the inability to make food choices freely, such as financial issues to purchase certain food items, or social influences that would lead individuals to contextual barriers.

Thank you for the suggestion. I expanded the discussion in the introductory paragraph to include a broader range of factors that influence food preferences, such as animal welfare, environmental impact, religious beliefs and personal aversions to meat. I also discussed the social and economic barriers that limit freedom of food choice.

How did you measure physical activity?

Thank you for your request for clarification. Physical activity was assessed by means of a structured questionnaire collecting information on the type of sport practised and the amount of weekly hours devoted to physical activity. Participants were classified into three predefined categories: less than 5 hours per week, between 5 and 10 hours, and more than 10 hours per week of physical activity. In addition, the data were discussed with a dietitian experienced in sports nutrition, who also has trainer skills, to validate the consistency of the responses and investigate any discrepancies. I have updated the Methods section to reflect this procedure.

Could you provide a body shape index (BSI) along with BMI a metric for assessing the health implications of a given human body height, mass and waist circumference?

Thank you for the suggestion. I calculated the BSI for all participants and included it in the Methods, Results and Table 1 to improve the assessment of body composition. The BSI is a parameter that integrates BMI, height and waist circumference to provide a more accurate measure of cardiometabolic risk, overcoming some of the limitations of BMI alone. In the method, I described the calculation of the BSI using the formula proposed by Krakauer and Krakauer, which considers waist circumference in metres divided by BMI raised by two-thirds and height raised by one-half. I also emphasised the value of this indicator in determining metabolic risk independently of total body mass. In the results, I reported the average BSI values in the sample, showing statistically significant differences between men and women. The results show that the BSI is higher in men than in women, in line with their greater predisposition to abdominal fat accumulation. However, the effect size is small, indicating limited variability between groups.In Table 1, I have added the BSI alongside the other anthropometric metrics, with a reference to the formula used in the table caption. Thanks again for the suggestion, which helped to improve the completeness of the analysis.

Round 2

Reviewer 1 Report

Comments and Suggestions for Authors

It is difficult for me to say that the author's revision addresses the current study's limitations, which still exist. Therefore, the value of the research in the present manuscript is limited, and the helpful reference for researchers in related fields is of little significance. Thus, the current revision cannot be applied to Foods publication, which will lead readers to question the credibility of the journal.  

Comments on the Quality of English Language

The problem of research content is more important than the language.

Author Response

Dear reviewer,

Thank you for your feedback. I acknowledge your concerns about the limitations of the study, its scientific value and its relevance to researchers in related fields. I have undertaken substantial revisions to clarify the study's contributions and address methodological concerns.

The introduction has been reduced to the essentials, while meeting the demands of other reviewers. I have added some results as requested by the editor on the consumption of other vegetable protein sources (cereals and nuts).  I have also completely rewritten the discussion to make it more incisive and tried to highlight differences in my data with recent literature. The practical implications of these findings for public health and nutrition policy are reinforced. The conclusions are now more original in highlighting the key findings of the study and their implications for dietary strategies that address gender and socioeconomic disparities.

The limitations are now discussed explicitly in the main text, with a more detailed breakdown in Table S1 of the Supplementary Material. I acknowledged the potential biases associated with self-reported dietary data, including recall and social desirability bias, and specified the strategies used to mitigate them. The cross-sectional design was further discussed, emphasising that causal relationships cannot be inferred. Furthermore, the limitations of BIA analysis and the role of unmeasured psychological and behavioural factors in determining protein choices were clarified.

I appreciate your critical review, which allowed me to improve the clarity, rigour and relevance of the manuscript. I trust that these revisions have strengthened the study and that I welcome further comments.

Kind regards,

The author